# TCMReasonSet: A Dataset for Explainable Medical Reasoning in Traditional Chinese Medicine

## Abstract

Large language models (LLMs) excel in structured tasks such as mathematics and programming but remain limited in knowledge-intensive domains like Traditional Chinese Medicine (TCM), which require complex reasoning. The primary bottleneck stems from the scarcity of high-quality training corpora that are well-structured and explicitly traceable in their reasoning pathways. To address this, we introduce TCMReasonSet, a high-quality dataset specifically designed for TCM clinical reasoning, aimed at enhancing the reliability and interpretability of LLMs in solving TCM-related problems. The construction of TCMReasonSet comprises three core components: (1) a proprietary TCM knowledge graph we developed — containing 52,000 entities and 1.38 million relations — serving as the foundation for dynamic retrieval and reasoning; (2) the generation of clinical question-answer pairs using LLMs, grounded in the aforementioned knowledge graph; and (3) building upon the knowledge graph and QA pairs, we propose the "TCM Tree-of-Thought" (TCM-ToT) methodology, which incorporates a dual-dimension scoring mechanism (logical consistency + factual accuracy) to evaluate clinical QA pairs and transform them into coherent, interpretable reasoning chains with explicit pathways. Through this pipeline, we ultimately generated 36,573 clinically interpretable reasoning samples. Experimental results demonstrate that fine-tuning models with TCMReasonSet significantly enhances medical problem-solving performance: the DeepSeek-Distill-8B model achieves an 8.9% accuracy gain, while our TCMReason-8B model surpasses the current state-of-the-art medical reasoning model by a 5.7% margin. Furthermore, expert evaluations further validate the reliability of our dataset in terms of factual accuracy and logical coherence.

## 1 Introduction

In recent years, large language models (LLMs) have achieved remarkable success in tasks such as mathematical reasoning, logical inference, and code generation Zhang et al. (2024); Pei et al. (2025); Zhao et al. (2023). Among these advancements, the Chain-of-Thought (CoT) paradigm has proven particularly effective across a wide range of reasoning tasks Feng et al. (2023); Liu et al. (2024). However, in knowledge-intensive domains like Traditional Chinese Medicine (TCM), the reasoning capabilities of LLMs remain limited and face substantial challenges Liévin et al. (2024); Yang et al. (2024b).

The primary challenge lies in the severe scarcity of high-quality, domain-adapted reasoning data. Unlike Western medicine, which is grounded in standardized systems, traditional Chinese medicine (TCM) emphasizes holistic perspectives, dynamic pattern differentiation, and the transmission of experiential knowledge—resulting in a knowledge structure that is highly complex and context-dependent. At present, there is a lack of publicly available datasets that are authoritative, logically coherent, and capable of supporting interpretable multi-step reasoning, which severely hampers the development of reasoning capabilities in large language models (LLMs) within the TCM domain.

Another critical limitation arises when attempting to construct TCM reasoning data using existing techniques. Current mainstream chain-of-thought (CoT) paradigms are ill-suited to the distinctive

characteristics of TCM diagnosis and treatment. The linearity and autoregressive nature of CoT reasoning fail to capture the multidimensional, nonlinear, and dialectical logic inherent in TCM practice—such as the principles of "treating the same disease with different therapies" and "treating different diseases with the same therapy" Wang et al. (2025a). This mismatch between the reasoning approach and the characteristics of the TCM system not only degrades reasoning performance but also risks the progressive accumulation and propagation of errors throughout the reasoning process—a particularly hazardous outcome in clinical applications.

To address these challenges, we introduce TCMReasonSet—the first large-scale, open-source reasoning dataset tailored for traditional Chinese medicine (TCM), designed to support high-quality and interpretable medical reasoning. The construction of this dataset proceeds in three stages:

- Stage 1: Knowledge-guided foundation. We construct a TCM knowledge graph (TCM-KG) containing 52,000 entities and 1.38 million relations, which provides a factual backbone for subsequent reasoning tasks.

- Stage 2: Construction of TCM-specific QA pairs. Building upon this graph, we leverage large language models (LLMs) to generate 50,000 knowledge-constrained, multitask QA pairs covering TCM theory, diagnosis, clinical practice, and pharmacology.

- Stage 3: Integration of domain knowledge into Tree-of-Thought (ToT) reasoning. Given the high-stakes nature of medical applications, each reasoning step must be verifiable and firmly grounded in expert knowledge Wen et al. (2023). Inspired by the RATT framework Zhang et al. (2025a), we design a novel data generation pipeline that deeply integrates TCM-KG into the ToT reasoning paradigm, ultimately producing more than 30,000 high-quality reasoning samples.

Our key contributions are threefold:

- Dataset release: Introduction of TCMReasonSet, the first large-scale, open-source reasoning dataset for TCM domain, consisting of 30,000 high-quality and interpretable reasoning samples.

- Knowledge-constrained generation: Development of a knowledge-constrained data generation pipeline grounded in a TCM knowledge graph, which substantially improves the interpretability and clinical reliability of TCM reasoning data.

- Empirical validation: Comprehensive experiments show that models fine-tuned on TCMReasonSet achieve substantial performance gains across multiple TCM benchmark suites. Notably, TCMReason-8B, fine-tuned on this dataset, establishes state-of-the-art results among medical models with fewer than 10 billion parameters.

## 2 RELATED WORK

### 2.1 REASONING DATA FOR MODEL ENHANCEMENT

In recent years, LLMs have demonstrated remarkable reasoning capabilities in domains such as mathematics and programming Satpute et al. (2024); Luo et al. (2023); Cai et al. (2023), motivating interest in applying similar methodologies to professional fields like Traditional Chinese Medicine (TCM). However, training high-quality reasoning models typically requires large-scale datasets annotated with fine-grained reasoning steps—resources that are prohibitively expensive to curate manually Zhao et al. (2025). A common alternative is to distill knowledge from more powerful LLMs Zhang et al. (2023). Yet, in knowledge-intensive tasks, this approach is prone to hallucination—the generation of factually incorrect or unverifiable content—a limitation that is especially pronounced in the TCM domain Zhang et al. (2025b); Xu et al. (2023). To address this issue, we introduce the TCMReasonSet dataset, which centers on structured Traditional Chinese Medicine Thought Trees (TCM-ToT). This dataset is designed to guide LLMs toward generating more reliable, fact-grounded, and interpretable reasoning chains for TCM tasks. By incorporating domain-specific knowledge and explicit reasoning structures, TCMReason provides a foundation for enhancing intelligent decision support systems in TCM with transparent and trustworthy outputs.

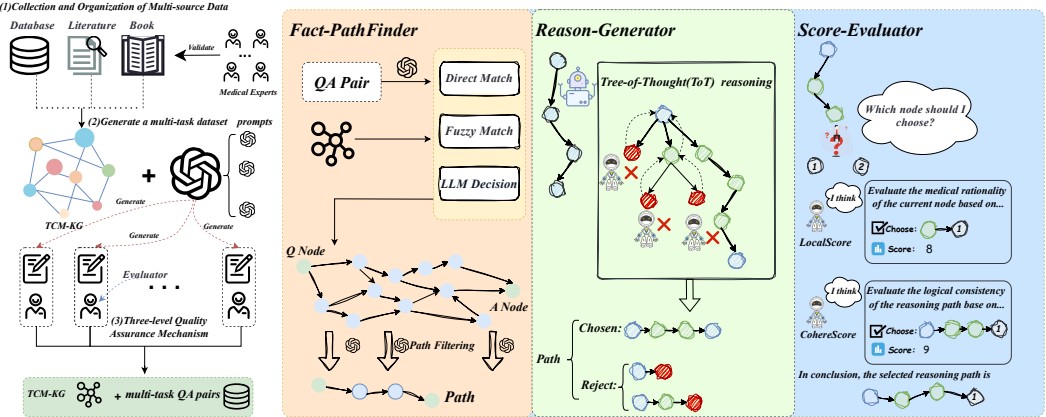

Figure 1: Overview of Our Data Generation Pipeline. The pipeline comprises three stages: (1) construction of a Traditional Chinese Medicine knowledge graph (TCM-KG) and automated generation of multi-task question-answer (QA) pairs; (2) identification of factual reasoning paths via entity linking and constrained path retrieval; and (3) synthesis of Tree-of-Thought (ToT) reasoning instances grounded in the extracted paths

## 2.2 THOUGHT STRUCTURES FOR LLMS

Structured reasoning paradigms, as advanced prompt engineering techniques, have shown strong potential in enhancing the logical coherence and interpretability of large language model (LLM) outputs Minaee et al. (2024); Zhang et al. (2022). Among them, the Chain-of-Thought (CoT) approach improves the model's reasoning capacity by explicitly decomposing tasks into intermediate logical steps Wei et al. (2022). Building upon this, the Tree-of-Thought (ToT) framework introduces a branching search process Yao et al. (2023), which enables forward planning, hypothesis exploration, and backtracking-based error correction, thereby demonstrating superior performance on cognitively demanding reasoning tasks Ni et al. (2025); Yang et al. (2025b). Given the inherent complexity and non-linear diagnostic reasoning in Traditional Chinese Medicine (TCM), this study systematically adapts and extends the ToT paradigm to TCM knowledge modeling. We propose TCM-ToT, a fact-guided reasoning method that tightly integrates the TCM knowledge graph into the reasoning process. It dynamically adjusts reasoning trajectories based on a dual-dimensional scoring mechanism—evaluating both logical coherence and factual accuracy. This method enables the generation of highly interpretable and reliable reasoning paths, supporting key applications in TCM decision-making, question answering, and educational scenarios.

## 3 PROPOSED METHOD

This section introduces TCMReason—a knowledge-guided reasoning data generation pipeline tailored for Traditional Chinese Medicine (TCM). As illustrated in Figure 1, the pipeline comprises four core modules that sequentially transform a knowledge graph into high-quality reasoning samples: (1) the QA-Pair Generator constructs(QA-Pair-Generator) a TCM knowledge graph by integrating multi-source data and then generates multi-task TCM question-answer pairs based on this graph; (2) the Fact-Path Finder(Fact-PathFinder)leverages structured medical knowledge from the TCM domain knowledge graph to construct fact-guided reasoning paths; (3) the Reason Generator(Reason-Generator)traverses entities related to the extracted paths to build interpretable reasoning chains; and (4) the Score Evaluator(Score-Evaluator)employs a dual-dimensional assessment mechanism that jointly evaluates factual accuracy and logical consistency to filter generated reasoning chains for validity and coherence.

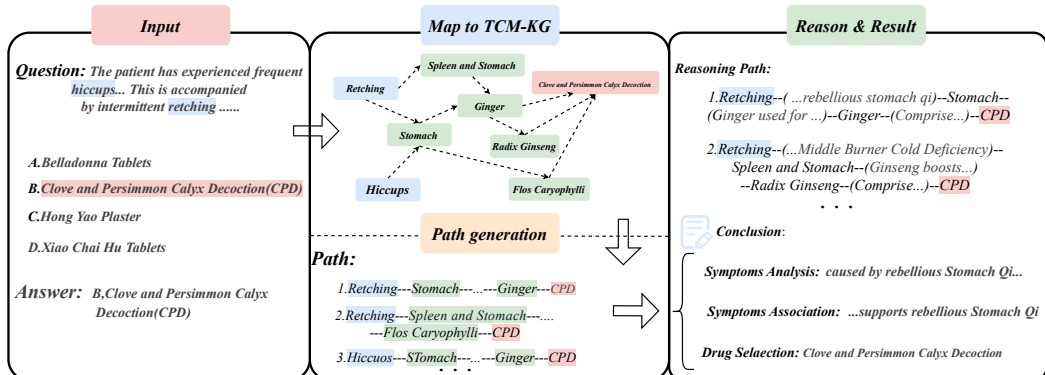

Figure 2: This example illustrates the end-to-end workflow of the TCMReason data generation pipeline. The figure depicts the full pipeline, beginning with a user query, followed by medical entity extraction via a large language model (LLM), entity alignment through knowledge graph mapping, and reasoning path generation with subsequent semantic pruning. Next, Tree-of-Thought (ToT) reasoning is performed along the selected paths, guided by a dual-dimensional scoring mechanism to ensure both logical coherence and factual accuracy.

## 3.1 QA-PAIR-GENERATOR

We construct the largest and most comprehensive Traditional Chinese Medicine (TCM) knowledge graph by integrating multi-source heterogeneous data into a high-quality, semantically rich structure. Our sources include: (1) structured medical entities from over ten authoritative TCM databases (e.g., TCMIO, ITCM, SymMap, TARKG); (2) structured knowledge extracted from more than 2,000 pharmacological research papers validated by domain experts; and (3) official drug standards from the Pharmacopoeia of the People's Republic of China and over 5,000 real-world classical clinical case records. After rigorous preprocessing—data cleaning, entity normalization, and semantic alignment—we built a high-quality TCM knowledge graph containing 52,000 entities (e.g., herbal formulas, syndromes, Chinese medicinal herbs, active compounds, molecular targets) and approximately 1.38 million semantic relations (e.g., formula–syndrome associations, compound–target interactions). Leveraging this graph, we propose a prompt engineering–based method to automatically generate diverse question-answering (QA) pairs, forming a multi-task TCM QA dataset covering core tasks such as fundamental TCM knowledge querying, classical case analysis, syndrome–disease reasoning, and herb/formula recommendations. To ensure logical coherence and factual accuracy, we implemented a three-tier quality assurance framework: (1) Automated evaluation: a domain-adapted LLM scorer retained only QA pairs scoring ¿90/100 for logical consistency and factual correctness; (2) Human validation: 1,000 random samples per task underwent dual blind annotation, achieving ¿90% inter-annotator agreement; (3) Expert review: ten senior TCM experts independently rated a sample of 100 QA pairs, yielding an average score ¿90/100, confirming clinical and academic reliability.

## 3.2 FACT-PATHFINDER

Fact-PathFinder leverages structured knowledge graphs to generate fact-guided reasoning paths, serving as the foundation for informed and interpretable decision-making.

Given an input question–answer pair $(Q, A)$, we employ a large language model (LLM) to perform semantic parsing and extract the associated sets of medical entities, denoted as $\{e_i^Q\}_{i=1}^{n_Q}$ and $\{e_j^A\}_{j=1}^{n_A}$. These entities are then mapped to nodes in the TCM-specific knowledge graph via a hierarchical three-stage matching strategy. First, exact string matching is applied. If no match is found, the system performs a vector-based similarity search and selects the top-1 candidate whose cosine similarity exceeds a predefined threshold ($\alpha = 0.85$). When both prior strategies fail, an LLM-driven semantic decision mechanism is activated, which selects the most contextually appropriate

candidate node by jointly analyzing the semantic content of $Q$ and $A$. This yields aligned entity sets $\{\hat{e}_i^Q\}_{i=1}^{n_Q}$ and $\{\hat{e}_j^A\}_{j=1}^{n_A}$.

For each aligned entity pair $(\hat{e}_i^Q, \hat{e}_j^A)$, the system retrieves all shortest paths connecting them in the knowledge graph, forming a preliminary path set $\tilde{P}_{ij}$. This ensures minimal path complexity to support interpretability. To refine the collection of all candidate paths $\bigcup_{i,j} \tilde{P}_{ij}$, we introduce an LLM-guided semantic pruning mechanism. Conditioned on the clinical context provided by $Q$, the LLM filters out irrelevant or semantically weak paths and selects the global top-$k$ most contextually salient reasoning chains (e.g., "lower back and knee weakness $\rightarrow$ kidney deficiency $\rightarrow$ *Rehmannia* $\rightarrow$ *Acanthopanax*").

Finally, we construct a global reasoning path set $\mathcal{P} = \{p_i\}_{i=1}^k$, where each path $p_i = (v_1^{(i)}, v_2^{(i)}, \ldots, v_{n_i}^{(i)})$ denotes a directed sequence of knowledge graph nodes of variable length. This set $\mathcal{P}$ serves as the foundational input to the subsequent Tree-of-Thought (TCM-ToT) reasoning module.

## 3.3 REASON-GENERATOR

Reason-Generator employs a knowledge graph-driven tree-of-thought (KG-ToT) mechanism to construct hierarchical reasoning paths, enabling interpretable and domain-aligned inference tailored to Traditional Chinese Medicine (TCM) decision-making.

Given the set of fact-guided reasoning paths $\mathcal{P} = \{p_i\}_{i=1}^k$ generated by Fact-PathFinder, where each path $p_i$ consists of a sequence of entity nodes from the TCM-specific knowledge graph (TCM-KG)—such as etiology, pathogenesis, symptoms, and treatments—we enhance each path with natural language explanations. Each $p_i$ represents a semantically coherent medical fact path grounded in clinical reasoning. To improve interpretability, we incorporate a large language model (LLM)-based explanatory mechanism that generates intermediate reasoning nodes between every pair of adjacent entity nodes, thereby enriching the symbolic path with human-readable justifications.

The resulting structure forms a reasoning tree $\mathcal{T}_{\text{KG}}$, which integrates both graph-anchored medical facts and LLM-generated explanatory narratives, serving as the foundation for transparent and clinically plausible inference.

The above process is formalized as follows:

$$\text{Branch}(p_i) = \big(\text{PathNode}(v_1^{(i)}), \text{ReasonNode}_1^{(i)}, \text{PathNode}(v_2^{(i)}), \text{ReasonNode}_2^{(i)}, \\ \ldots, \text{PathNode}(v_{n_i}^{(i)})\big) \tag{1}$$

$$\text{ReasonNode}_j^{(i)} = \text{ToT}(v_j^{(i)}, v_{j+1}^{(i)}), \quad \text{for } j = 1, \ldots, n_i - 1 \tag{2}$$

$$\mathcal{T}_{\text{KG}} = \bigcup_{p_i \in \mathcal{P}} \text{Branch}(p_i) \tag{3}$$

where $\text{PathNode}(v_j^{(i)})$ denotes an entity node inherited from the knowledge graph, serving as an anchor for medical facts such as pathology, symptoms, or treatment plans. $\text{ReasonNode}_j^{(i)} = \text{ToT}(v_j^{(i)}, v_{j+1}^{(i)})$ represents an intermediate node produced by the LLM, encoding causal, pathological, or TCM-specific reasoning (e.g., syndrome differentiation) between $v_j^{(i)}$ and $v_{j+1}^{(i)}$.

### 3.3.1 REASONING BETWEEN NODES AND PATH EXPLANATION

After constructing the set of reasoning paths $\mathcal{P} = \{p_i\}_{i=1}^k$, the system sequentially invokes the large language model to perform Tree-of-Thought (ToT) reasoning over each adjacent node pair $(v_i^{(i)}, v_{i+1}^{(i)})$ within each path. For every such pair, the LLM not only infers a reasoning outcome but also produces a coherent chain of thoughts that explicates the underlying causal relationships and domain-specific medical logic—such as syndrome differentiation, pathogenesis progression, or therapeutic rationale—linking the two nodes.

$$\text{ToT}_{\text{LLM}}(v_i^{(i)}, v_{i+1}^{(i)}) \Rightarrow \text{Reasoning}(v_j^{(i)} \rightarrow v_{j+1}^{(i)}) \tag{4}$$

where $(v_i^{(i)}, v_{i+1}^{(i)})$ represents the node pairs of the reasoning input (e.g., syndrome types, Eight Principles characteristics, medicinal properties and their meridian associations, etc.). $\text{Reasoning}(v_j^{(i)} \rightarrow v_{j+1}^{(i)})$ represents the reasoning output, including the result node and a structured or natural language-based medical reasoning explanation.

### 3.3.2 DYNAMIC BACKTRACKING MECHANISM

If a segment of the ToT reasoning output is deemed invalid by the scoring mechanism, the system triggers a backtracking procedure to the most recent verified knowledge node (i.e., a PathNode). From this anchor point, alternative reasoning trajectories are regenerated to preserve logical coherence and clinical correctness.

The backtracking target is formally defined as:

$$v_{\text{backtrack}}^{(i)} = \arg \max_{n \in \text{Ancestors}(v_{\text{err}}^{(i)})} \text{depth}(v) \quad \text{if } \mathbb{I}[\text{Type}(v) = \text{PathNode}] \tag{5}$$

where $\text{Ancestors}(v_{\text{err}}^{(i)})$ represents all ancestor nodes pointing to the current erroneous node. $v_{\text{backtrack}}^{(i)}$ represents the most recent knowledge anchor point returned by the system, and then re-expands other potential reasoning paths.

### 3.4 SCORE-EVALUATOR

Traditional methods for generating reasoning paths over knowledge graphs typically rely on structured algorithms, such as shortest path search or random walks. However, these methods struggle to fully capture the unique diagnostic and treatment patterns of Traditional Chinese Medicine (TCM) and the domain-specific safety constraints. To ensure both the accuracy and consistency of tree-structured reasoning grounded in knowledge graphs for medical tasks, we propose an innovative dual-dimensional scoring mechanism. This mechanism performs semantic-level evaluations at both the local (node-level, i.e., single-step reasoning) and global (path-level, i.e., multi-step reasoning) scales. The mechanism includes two core components: (1) **accuracy of single-step facts**: verifying the medical validity of inferences between adjacent nodes; (2) **rationality of multi-step logic**: examining the logical consistency of the entire reasoning path. To implement this scoring mechanism, we employ the medical-domain fine-tuned large language model "HuatuoGPT-o1-72B" **?** as the scoring oracle.

### 3.4.1 ACCURACY OF SINGLE-STEP FACTS

This component evaluates the medical validity of each reasoning step in context, detecting issues such as contraindicated drug interactions, logical inconsistencies, and contradictions in TCM syndrome differentiation.

$$\text{LocalScore} = \text{HuaTuo}_{\theta}(I_{\text{local}}, x_{\text{node}}) \in [0, 10] \tag{6}$$

where $I_{\text{local}}$ denotes the instruction template for local accuracy evaluation, and $x_{\text{node}}$ represents the current node information (including entity type and attributes).

### 3.4.2 RATIONALITY OF MULTI-STEP LOGIC

This component assesses the logical coherence and medical plausibility of pathogenesis evolution along the entire reasoning path, ensuring adherence to established clinical inference principles.

$$\text{CohereScore} = \text{HuaTuo}_{\theta}(I_{\text{cohere}}, x_{\text{path}}) \in [0, 10] \tag{7}$$

where $I_{\text{cohere}}$ denotes the instruction template for path-level coherence evaluation, and $x_{\text{path}}$ represents the sequence of nodes and their semantic relationships along the reasoning path.

# 4 EXPERIMENT

## 4.1 EXPERIMENTAL SETUP

### 4.1.1 REASONING DATA GENERATION AND PREPROCESSING

Based on a rigorously validated, high-quality knowledge graph (TCM-KG), we developed a reasoning-oriented dataset using the TCMReason data generation pipeline. The resulting dataset, named **TCMReasonSet**, comprises over 30,000 Tree-of-Thought (ToT) reasoning samples spanning a broad range of tasks in Traditional Chinese Medicine (TCM). Each sample includes a structured and interpretable reasoning chain. TCMReasonSet is designed to serve as an efficient and scalable resource for TCM-specific reasoning and model training, thereby enhancing the reasoning capabilities of large language models and improving the transparency of clinical decision-making.

### 4.1.2 BENCHMARK DATASETS

We rigorously evaluate the model's performance using standardized benchmarks datasets in TCM: the Chinese Medicine Benchmark (CMB) (Wang et al., 2024), the Traditional Chinese Medicine Standardized Diagnostic Test (TCM-SDT) (Wang et al., 2025b), the Medical Language Evaluation Corpus (MLEC-QA) (Li et al., 2021), and the TCM Licensing Examination Database (TCM-exam) (SylvanL, 2024). CMB, as a comprehensive evaluation benchmark dedicated to the field of Chinese medicine, categorizes the TCM domain evaluation data into four subsets: (1) Chinese Materia Medica (CHM), focusing on the properties of medicinal materials, compatibility principles, and the application of formulas; (2) TCM Diagnosis (TCM-DS), emphasizing clinical differentiation and reasoning abilities; (3) Basic TCM Theory (BTT), covering core theories such as Yin-Yang and the Five Elements; and (4) Postgraduate Entrance Exam Questions (PEEQ), encompassing advanced medical education-related questions. TCM-SDT specializes in TCM differentiation decision-making through patient simulations, MLEC-QA evaluates the performance of medical question-answering systems in complex reasoning tasks, and TCM-exam assesses qualifications for TCM practitioners.

### 4.1.3 COMPARISON MODELS

To systematically evaluate the generalization performance of the TCMReasonSet across different base models, we select several models with fewer than 10 billion parameters as baselines and perform supervised fine-tuning (SFT) on the dataset. Specifically, we focus on two representative instruction-tuned models: Qwen2.5-7B-Instruct (Yang et al., 2024a) and Mistral-Instruct-7B (Jiang et al., 2023). Based on the experimental framework by Chen et al. (2024b), we employ a learning rate of 5e-6, a batch size of 128, and utilize DeepSpeed-ZeRO stage 3 optimization techniques (Rajbhandari et al., 2020) to complete three epochs of training. To further investigate the impact of the dataset on reasoning ability, we also fine-tune Qwen3-8B (Yang et al., 2025a), DeepSeek-Distill-8B (Guo et al., 2025), and Huatuo-o1-RL-8B (Chen et al., 2024a) using the same hyperparameters. Finally, as shown in Table 3, we compare the fine-tuned models against three categories of baselines: (1) General-purpose large language models, including LLaMA 3.1-Instruct-8B (Dubey et al., 2024), Mistral-Instruct-7B, and Qwen-Instruct-7B; (2) Medical domain-specific models, such as BianQue (Chen et al., 2023), Bentsan (Wang et al., 2023), BianCang (Wei et al., 2024), ShengNong-TCM (Wei Zhu & Wang, 2023), and Taiyi (Luo et al., 2024); and (3) Medical reasoning optimization models, including Medical-CoT (Mahmoud, 2025), Deepseek-Distill-8B and Huatuo-o1-RL.

## 4.2 EXPERIMENT RESULTS

We follow the experimental design methodology of MedReason (Wu et al., 2025) to comprehensively evaluate the performance of the proposed framework from two perspectives: data validity and model validity. (1) **Data Validity Evaluation**: To assess the quality and utility of the constructed dataset, we conduct instruction tuning on LLMs using both our proposed dataset and the Huatuo-CoT dataset. In addition, we fine-tune a dedicated reasoning model exclusively on our dataset. This setup enables a comparative analysis of the impact of different data sources on instruction-following and reasoning capabilities. (2) **Model Validity Evaluation**: We evaluate the model fine-tuned with our dataset against several state-of-the-art models in the domain of Traditional Chinese Medicine

| Benchmarks | Qwen2.5-7B-Instruct | | | Mistral-Instruct-7B | | |
|---|---|---|---|---|---|---|
| | original | w/ huatuo CoT | w/ ours | original | w/ huatuo CoT | w/ ours |
| TCM-exam | 76.1 | 80.4 (+4.3) | 82.2 (+6.1) | 38.8 | 41.3 (+2.5) | 47.7 (+8.9) |
| TCM-SDT | 52.9 | 57.8 (+4.9) | 61.5 (+8.6) | 23.3 | 31.7 (+8.4) | 33.9 (+10.6) |
| MLEC(A2) | 78.7 | 81.3 (+2.6) | 85.6 (+6.9) | 43.7 | 52.5 (+8.8) | 54.2 (+10.5) |
| MLEC(A3A4) | 75.2 | 78.9 (+3.7) | 80.5 (+5.3) | 41.3 | 43.6 (+2.3) | 44.8 (+3.5) |
| CMB-CHM | 59.1 | 63.3 (+4.2) | 67.4 (+8.3) | 35.5 | 41.2 (+5.7) | 44.3 (+8.8) |
| CMB-DS | 55.1 | 61.9 (+6.8) | 62.1 (+7.0) | 34.3 | 41.8 (+7.5) | 43.1 (+8.8) |
| CMB-BTT | 55.7 | 56.3 (+0.6) | 57.8 (+2.1) | 37.0 | 39.2 (+2.2) | 41.6 (+4.6) |
| CMB-PEEQ | 48.2 | 53.8 (+5.6) | 55.7 (+7.5) | 29.6 | 34.8 (+5.2) | 36.3 (+6.7) |
| Avg | 62.6 | 66.7 (+4.1) | **69.1 (+6.5)** | 35.4 | 40.7 (+5.3) | **43.2 (+7.8)** |

Table 1: Performance comparison of LLMs fine-tuned with Huatuo-CoT and TCMReasonSet (our dataset) under instruction-based fine-tuning.

| Base Model | Data | TCM-exam | TCM-SDT | MLEC(A2) | MLEC(A3A4) | Avg |
|---|---|---|---|---|---|---|
| Qwen3-8B | Original | 72.3 | 54.1 | 76.6 | 78.5 | 70.3 |
| | w/ ours | 78.4 (+6.1) | 58.9 (+4.8) | 82.5(+5.9) | 83.7 (+5.2) | 75.8(+5.5) |
| DeepSeek-Distill-8B | Original | 35.5 | 26.8 | 34.3 | 37.2 | 33.4 |
| | w/ ours | 42.9 (+7.4) | 34.5 (+7.7) | 44.1 (+9.8) | 45.3 (+8.1) | 41.4(+8.9) |

| Base Model | Data | CMB-CHM | CMB-DS | CMB-BTT | CMB-BEEQ | Avg |
|---|---|---|---|---|---|---|
| Qwen3-8B | Original | 56.4 | 57.6 | 58.3 | 47.8 | 55.0 |
| | w/ ours | 58.0 (+1.6) | 62.1 (+4.5) | 61.7 (+3.4) | 51.4 (+3.6) | 58.3(+3.3) |
| DeepSeek-Distill-8B | Original | 34.8 | 29.6 | 37.0 | 27.5 | 32.2 |
| | w/ ours | 37.5 (+2.7) | 34.2 (+4.6) | 42.5 (+5.5) | 35.6 (+8.1) | 37.4(+5.2) |

Table 2: Performance comparison of reasoning LLMs fine-tuned with TCMReasonSet (our dataset) versus the original models.

(TCM). The comparison is conducted across multiple benchmarks, focusing on metrics such as reasoning accuracy, interpretability, and alignment with domain knowledge. The results demonstrate that our framework significantly outperforms existing baselines in producing coherent, reliable, and clinically meaningful reasoning paths.

### 4.2.1 EVALUATION OF TCMREASONSET ON INSTRUCTION FINE-TUNED MODEL

In this section, we evaluate the effectiveness of instruction fine-tuning with TCMReasonSet as a data augmentation strategy. As shown in Table 1, we report the accuracy (%) of Qwen2.5-7B-Instruct and Mistral-Instruct-7B across multiple benchmark datasets in the TCM domain. Models fine-tuned on our TCMReasonSet (denoted as w/ours) consistently outperform both the base models and those fine-tuned on the Huatuo-CoT dataset across all evaluated benchmarks. Specifically, for Qwen2.5-7B-Instruct, fine-tuning with TCMReason yields an average accuracy improvement from 62.6% to 69.1% (+6.5%), exceeding the +4.1% gain achieved by the Huatuo-CoT-finetuned model. The improvement is even more pronounced for Mistral-Instruct-7B, where accuracy rises from 35.4% to 43.2% (+7.8%), compared to a +5.3% increase from Huatuo-CoT. These results confirm the effectiveness of our TCMReason data in enhancing the reasoning ability of instruction-tuned models, and demonstrate its superiority over existing domain-specific datasets in TCM.

### 4.2.2 EVALUATION OF TCMREASONSET ON REASONING MODELS

We further investigate the effect of fine-tuning LLMs using the TCMReasonSet to enhance their reasoning capabilities. As shown in Table 2, incorporating our dataset (denoted as w/ours) leads to substantial performance gains across multiple TCM question-answering benchmarks. Specifically, on the TCM-exam, TCM-SDT, and MLEC datasets, Qwen-8B achieves an average improvement of 5.5%, while DeepSeek-Distill-8B demonstrates a more pronounced gain of 8.9%. Similarly, on the CMB benchmark, Qwen-8B improves by 3.3%, and DeepSeek-Distill-8B shows an improvement of 5.2%. These results highlight the effectiveness of the TCMReason dataset—which integrates knowledge graph (KG)-based factual guidance with Tree-of-Thought (ToT) reasoning—in improving the reasoning performance of reasoning large language models within the TCM domain.

| Model | TCM-exam | TCM-SDT | MLEC(A2) | MLEC(A3A4) | CMB(TCM) | Avg |
|---|---|---|---|---|---|---|
| Llama3.1-Instruct-8B | 41.2 | 38.6 | 46.3 | 48.7 | 37.6 | 42.4 |
| Qwen2.5-Instruct-7B | 76.1 | 52.9 | 78.7 | 75.2 | 54.5 | 67.4 |
| Mistral-Instruct-7B | 38.8 | 23.3 | 43.7 | 41.3 | 34.1 | 36.2 |
| BianQue | 23.6 | 19.3 | 21.2 | 23.3 | 18.8 | 21.2 |
| Bnetsan | 29.2 | 17.3 | 22.6 | 23.0 | 20.3 | 22.4 |
| BianCang | 86.7 | 52.9 | 83.1 | 84.5 | 60.2 | 73.4 |
| ShengNong-TCM | 35.4 | 18.7 | 21.5 | 23.3 | 18.7 | 23.5 |
| Taiyi | 46.6 | 33.5 | 43.0 | 46.8 | 24.5 | 38.8 |
| Qwen3-8B | 72.3 | 54.1 | 76.6 | 78.5 | 55.0 | 67.3 |
| Medical-CoT-8B | 39.3 | 27.8 | 41.7 | 38.2 | 31.7 | 35.7 |
| DeepSeek-Distill-8B | 35.5 | 26.8 | 34.3 | 37.2 | 32.2 | 33.2 |
| Huatuo-o1-RL-8B | 85.4 | 60.6 | 82.7 | 85.3 | 57.1 | 74.2 |
| **TCMReason-8B (ours)** | **89.7** | **67.8** | **87.1** | **89.2** | **65.8** | **79.9** |

Table 3: Compare the TCMReason-8B model, trained on TCMReasonSet, against general-purpose large language models (LLMs) and medical-domain LLMs of comparable parameter scale across multiple Traditional Chinese Medicine (TCM) medical benchmarks.

| Dual-Dim Scoring | TCM-exam | TCM-SDT | MLEC(A2) | MLEC(A3A4) | CMB(TCM) | Avg |
|---|---|---|---|---|---|---|
| w/o Single-step Scoring | 75.2 | 49.6 | 76.3 | 75.1 | 52.8 | 65.8 |
| w/o Multi-step Scoring | 73.8 | 46.9 | 74.1 | 73.5 | 50.2 | 63.7 |
| w/ | **76.1** | **52.9** | **78.7** | **75.2** | **54.5** | **67.4** |

Table 4: Ablation study of the dual-dimensional scoring mechanism.

### 4.2.3 COMPARISON OF TCMREASON-8B AND EXISTING SOTA MODELS

We further fine-tuned the Huatuo-o1-RL-8B model using TCMReasonSet to develop a new TCM-specific reasoning model—TCMReason-8B. As shown in Table 3, TCMReason-8B consistently outperforms all baseline models across the four benchmark datasets, achieving an overall average score of 79.9%, which represents a 5.7% absolute improvement over the base Huatuo-o1-RL-8B. Notably, the model exhibits substantial gains on tasks requiring advanced diagnostic reasoning and syndrome differentiation. On the TCM-SDT benchmark, TCMReason-8B achieves a 7.8% improvement over the base model and surpasses all other baselines by more than 10% in several sub-tasks. These results strongly validate the effectiveness of our TCMReasonSet dataset, and the superior performance of TCMReason-8B establishes it as a new benchmark for reasoning capabilities in TCM-oriented artificial intelligence systems.

### 4.3 ABLATION STUDY

To assess the contribution of the proposed dual-dimensional scoring mechanism, we conducted an ablation study using the Qwen2.5-7B-Instruct model. As shown in Table 4, incorporating the dual-scoring mechanism leads to notable performance gains across most TCM benchmarks—yielding an average improvement of 1.6% over the multi-step scoring variant and 3.7% over the single-step variant. These results underscore the importance of the dual-scoring mechanism in enhancing the reasoning capabilities of LLMs in medical applications by ensuring higher-quality, semantically coherent training data. Additional details of the ablation setup and results are provided in the Appendix.

## 5 CONCLUSION

We prop we construct TCMReasonSet, a high-quality reasoning dataset to support model training in TCM diagnostic tasks.Experimental results show that models fine-tuned on our dataset achieve state-of-the-art performance across multiple TCM benchmarks, particularly in complex reasoning scenarios. This work demonstrates the effectiveness of combining domain knowledge with LLM-based reasoning, offering a scalable and interpretable approach for trustworthy medical AI in the TCM domain.

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

# A APPENDIX

## A.1 DATA DESCRIPTION

This chapter presents the knowledge graph data and the generated question-answer (QA) pairs. As shown in Figure 3, the Traditional Chinese Medicine Knowledge Graph (TCM-KG) is stored in the Neo4j graph database. figure 4 presents the multi-task QA pairs, covering TCM knowledge such as fundamental theories, herbal recommendations, prescription recommendations, and more, comprising over 50,000 high-quality data samples.

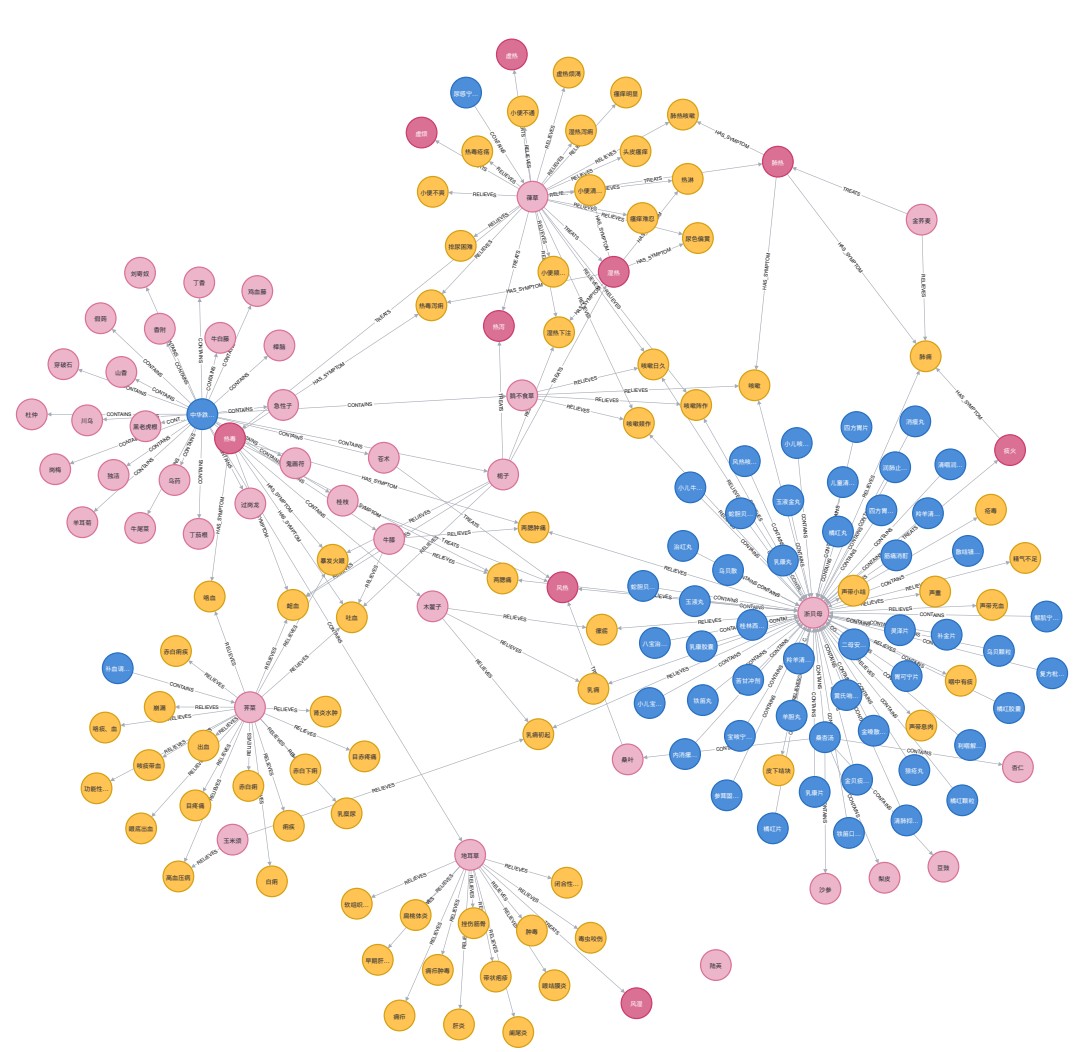

Figure 3: Illustration of the TCM Knowledge Graph (TCM-KG)

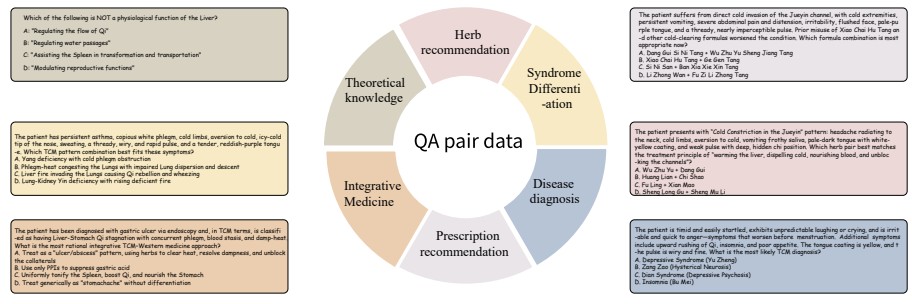

Figure 4: Detailed information on the multi-task dataset

| KG | TCM-exam | TCM-SDT | MLEC(A2) | MLEC(A3A4) | CMB(TCM) | Avg |
|---|---|---|---|---|---|---|
| W/o | 65.8 | 45.3 | 72.5 | 68.5 | 51.8 | 60.7 |
| W/ | 76.1 | 52.9 | 78.7 | 75.2 | 54.5 | 67.4 |

Table 5: Ablation study results: Impact of knowledge graph as factual guidance on TCM benchmarks.

| Method | Expert 1 | Expert 2 | Expert 3 | Expert 4 | Expert 5 | Avg |
|---|---|---|---|---|---|---|
| CoT | 6 | 7 | 5 | 7 | 6 | 6.2 |
| AoT | 6 | 6 | 7 | 6 | 8 | 6.6 |
| ToT | 7 | 6 | 7 | 7 | 8 | 7.0 |
| GoT | 7 | 8 | 7 | 8 | 8 | 7.6 |
| **TCM-ToT (Ours)** | **9** | **8** | **7** | **9** | **9** | **8.4** |

Table 6: Performance comparison of the dual-dimensional scoring-based TCM-ToT method with other reasoning strategies. The scores represent the average accuracy across multiple TCM benchmarks.

### A.2 ALGORITHM DETAILS

This section provides detailed pseudocode for the core components of our data generation pipeline, including the knowledge graph path discovery mechanism and the tree-of-thought reasoning with dual-dimensional scoring.

---

**Algorithm 1** Knowledge Graph Path Discovery for TCM Reasoning (Fact-PathFinder)

---

**Require:** Question–Answer pair $(Q, A)$, TCM Knowledge Graph $G$, similarity threshold $\alpha = 0.85$, path limit $k$, LLM function $f_L$

**Ensure:** Set of fact-guided reasoning paths $\mathcal{P} = \{p_i\}_{i=1}^{k}$

1: $\{e_i^Q\}_{i=1}^{n_Q} \leftarrow$ extract_entities$(Q, f_L)$ {Extract entities from question via semantic parsing}
2: $\{e_j^A\}_{j=1}^{n_A} \leftarrow$ extract_entities$(A, f_L)$ {Extract entities from answer via semantic parsing}
3: $\{\hat{e}_i^Q\}_{i=1}^{n_Q}, \{\hat{e}_j^A\}_{j=1}^{n_A} \leftarrow \emptyset, \emptyset$
4: **for** each entity $e \in \{e_i^Q\}_{i=1}^{n_Q} \cup \{e_j^A\}_{j=1}^{n_A}$ **do**
5:    $matched \leftarrow$ exact_match$(e, G)$ {Stage 1: Exact string matching}
6:    **if** $matched =$ None **then**
7:       $candidates \leftarrow$ vector_search$(e, G, \alpha)$ {Stage 2: Cosine similarity search $(\geq \alpha)$}
8:       **if** $|candidates| > 0$ **then**
9:          $matched \leftarrow$ top_candidate$(candidates)$ {Select top-1 candidate}
10:       **else**
11:          $matched \leftarrow f_L($semantic_select$, e, G, Q, A)$ {Stage 3: LLM-driven semantic selection using clinical context}
12:       **end if**
13:    **end if**
14:    Add $matched$ to corresponding $\{\hat{e}_i^Q\}$ or $\{\hat{e}_j^A\}$
15: **end for**
16: $\tilde{\mathcal{P}} \leftarrow \emptyset$ {Preliminary path set}
17: **for** each pair $(\hat{e}_i^Q, \hat{e}_j^A) \in \{\hat{e}_i^Q\}_{i=1}^{n_Q} \times \{\hat{e}_j^A\}_{j=1}^{n_A}$ **do**
18:    $paths_{ij} \leftarrow$ find_shortest_paths$(\hat{e}_i^Q, \hat{e}_j^A, G)$ {Retrieve all shortest paths for minimal complexity}
19:    $\tilde{\mathcal{P}} \leftarrow \tilde{\mathcal{P}} \cup paths_{ij}$
20: **end for**
21: $\mathcal{P} \leftarrow f_L($semantic_prune$, \tilde{\mathcal{P}}, Q, k)$ {LLM-guided pruning: retain top-$k$ contextually salient paths}
22: **return** $\mathcal{P}$

---

---

**Algorithm 2** Tree-of-Thought Reasoning with Dual-Dimensional Scoring

---

**Require:** Path set $\mathcal{P} = \{p_i\}_{i=1}^{k}$, LLM function $f_L$, scoring model $f_S$, score thresholds $\theta_{local}$, $\theta_{global}$

**Ensure:** Enhanced reasoning tree $\mathcal{T}_{final}$

1: $\mathcal{T}_{KG} \leftarrow \emptyset$ {Initialize reasoning tree}
2: **for** each path $p_i = (v_1^{(i)}, v_2^{(i)}, \ldots, v_{n_i}^{(i)}) \in \mathcal{P}$ **do**
3:   $branch_i \leftarrow [\text{PathNode}(v_1^{(i)})]$ {Initialize branch with first node}
4:   **for** $j = 1$ to $n_i - 1$ **do**
5:     $reasoning \leftarrow f_L(\text{ToT\_reason}, v_j^{(i)}, v_{j+1}^{(i)})$ {Generate reasoning}
6:     $local\_score \leftarrow f_S(\text{local\_eval}, reasoning, v_j^{(i)}, v_{j+1}^{(i)})$ {Local scoring}
7:     **if** $local\_score < \theta_{local}$ **then**
8:       $ancestor \leftarrow \text{find\_backtrack\_node}(branch_i)$ {Backtrack to last valid node}
9:       $reasoning \leftarrow f_L(\text{ToT\_reason}, ancestor, v_{j+1}^{(i)})$ {Regenerate reasoning}
10:       $local\_score \leftarrow f_S(\text{local\_eval}, reasoning, ancestor, v_{j+1}^{(i)})$
11:     **end if**
12:     Append ReasonNode($reasoning$) to $branch_i$
13:     Append PathNode($v_{j+1}^{(i)}$) to $branch_i$
14:   **end for**
15:   $global\_score \leftarrow f_S(\text{coherence\_eval}, branch_i)$ {Global coherence scoring}
16:   **if** $global\_score \geq \theta_{global}$ **then**
17:     $\mathcal{T}_{KG} \leftarrow \mathcal{T}_{KG} \cup \{branch_i\}$ {Accept high-quality branch}
18:   **end if**
19: **end for**
20: $\mathcal{T}_{final} \leftarrow \text{merge\_branches}(\mathcal{T}_{KG})$ {Construct final reasoning tree}
21: **return** $\mathcal{T}_{final}$

---

### A.3 ABLATION STUDY

To systematically evaluate the effectiveness and contribution of the proposed knowledge graph as factual guidance in the training of large language models, we designed and conducted a series of ablation experiments based on the Qwen2.5-7B-Instruct model. As shown in Table 5, the model variant incorporating the knowledge graph as external knowledge guidance consistently achieves significant performance improvements across multiple benchmark tasks in the field of traditional Chinese medicine (TCM). Compared to the baseline model without any factual guidance, it achieves an average accuracy improvement of 6.7%, with even greater gains observed in certain complex reasoning tasks. These results indicate that the knowledge graph not only provides structured domain-specific knowledge but also effectively enforces factual consistency during the training process, thereby mitigating hallucination phenomena and enhancing the model's understanding and reasoning capabilities regarding professional terminology, disease mechanisms, prescription compatibility, and other intricate medical logic. More importantly, as a reliable and trustworthy knowledge source, the knowledge graph facilitates the construction of more accurate and robust training samples, enabling the model to perform plausible inference based on structured facts when confronted with ambiguous or ill-defined inputs.

To further evaluate the performance of the proposed TCM-ToT method based on dual-dimensional scoring, we conducted comparative experiments against several mainstream reasoning frameworks: Chain-of-Thought (CoT), Algorithm-of-Thoughts (AoT), Tree-of-Thought (ToT), and Graph-of-Thought (GoT). For each task, ten questions were randomly selected, and reasoning paths were generated using the aforementioned methods. Subsequently, five domain experts in traditional Chinese medicine (TCM) were invited to conduct a 10-point scale manual evaluation of the generated outputs, based on three criteria: Safety, Professionalism, and Fluency. The results, summarized in Table 6, show that TCM-ToT significantly outperforms all baseline methods in terms of overall score. Specifically, it achieves an average improvement of 1.4 point over the original ToT approach and surpasses the second-best method, GoT, by 0.8 points. The expert evaluations consistently indicate that the reasoning paths generated by TCM-ToT are more logically rigorous, better supported

by medical evidence, and exhibit higher levels of professionalism and clinical interpretability. These findings validate the effectiveness and advantages of our proposed method in enhancing the reasoning quality of large language models within the domain of traditional Chinese medicine.

## A.4    EXPERT EVALUATION

We invited five domain experts in traditional Chinese medicine to conduct a comprehensive evaluation of the responses generated by TCMReason-8B, Taiyi, Qwen2.5-7B-Instruct, Huatuo-8B-RL, and BianCang, based on three dimensions: Safety, Professionalism, and Fluency. In the experiment, ten questions were randomly selected from the TCM-SDT benchmark dataset, and each model was prompted to reason and generate answers to these questions. The experts scored each model's performance based on the reasoning process and the quality of the final answer.

As shown in Figure  5, TCMReason-8B outperforms all other models across all three evaluation dimensions, demonstrating a comprehensive performance advantage.  Notably, TCMReason-8B achieves particularly outstanding results in the dimension of safety, significantly surpassing existing models. The experts consistently agreed that TCMReason-8B exhibits stronger factual accuracy and clinical compliance in its responses, effectively avoiding the generation of misleading or potentially risky content, thereby demonstrating higher safety and reliability in medical scenarios.

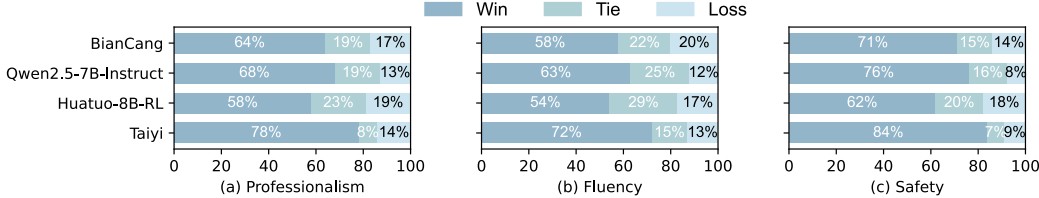

Figure 5: Experts' scoring results across three dimension

## A.5    MAIN PROMPTS USED IN THE FRAMEWORK

This section presents the key prompts employed in our TCMReason framework for different components of the reasoning pipeline.

## A.6    USE OF LARGE LANGUAGE MODELS (LLMS)

In accordance with ICLR's guidelines on the use of AI-assisted technologies, we declare that Large Language Models (LLMs) were used in the preparation of this manuscript for translation and language polishing purposes only. Specifically:

- **Translation**: LLMs were used to assist in translating portions of Chinese medical texts and terminology into English to ensure accurate representation of Traditional Chinese Medicine concepts.
- **Language polishing**: LLMs were employed to improve the grammatical accuracy, clarity, and readability of the manuscript text.

No LLMs were used in:

- The design or implementation of the proposed methodology
- The generation of experimental results or data analysis
- The formulation of research hypotheses or conclusions
- The creation of figures, tables, or other scientific content

All authors take full responsibility for the accuracy, originality, and scientific validity of the content presented in this paper.  The core research contributions, methodology, experimental design, and conclusions are entirely the work of the human authors.

864
865
866
867
868
869
870
871
872
873
874
875
876
877
878
879
880
881
882
883
884
885
886
887
888
889
890
891
892
893
894
895
896
897
898
899
900
901
902
903
904
905
906
907
908
909
910
911
912
913
914
915
916
917

**prompt for identifying in question and answer**

messages = [ {"role": "system", "content": "You are a TCM information processing assistant who is proficient in Traditional Chinese Medicine (TCM) theory and strictly follows instructions."}, {"role": "user", "content": """Within the context of TCM theory, clinical practice, or classical TCM literature, precisely extract all TCM-related entities from the given text.

Output Format:

Strictly adhere to the following JSON structure.

The type of each entity must belong exclusively to one of the following categories:

1. zhongyi_syndrome（中医证候，如"肝郁脾虚证""气虚血瘀证"）

2. disease（疾病名称，包括现代病名与中医病名，如"消渴""高血压"）

3. herb（中药饮片或药材，如"黄芪""当归"）

4. formula（方剂，如"四物汤""逍遥散"）

5. symptom（症状或体征，如"头晕""舌淡苔白""脉弦"）

6. meridian（经络，如"足少阳胆经""任脉"）

7. zangfu（脏腑，如"肝""脾""心肾不交"）

8. pathogenic_factor（病因病机，如"风寒""湿热""情志内伤"）

9. therapeutic_principle（治则治法，如"疏肝理气""健脾化湿""活血化瘀"）

10. acupuncture_point（穴位，如"足三里""太冲"）

```json
{
  "Entity": [
    {"id": "1", "type": "zhongyi_syndrome", "name": "肝郁脾虚证"},
    {"id": "2", "type": "herb", "name": "柴胡"}
  ]
```

Figure 6: Prompt for identifying entities in question and answer

**Prompt for generating answer with TOT reasoning**

messages = [ {"role": "system", "content": "You are an expert in the medical field, skilled at solving complex problems through systematic thinking."}, {"role": "user", "content": f"""Given a medical question, a set of initial reasoning paths, and a reference answer, your task is to simulate a Tree-of-Thought reasoning process: actively explore multiple possible reasoning branches, critically evaluate the plausibility of each path, and—based on medical knowledge—select or construct the optimal path to reach a conclusion.

##Core Requirements:
1.Generate multiple reasoning paths: Even if initial paths are provided, proactively consider
whether other plausible explanations or mechanisms exist. Treat all feasible paths as
branches of a "thought tree."
2.Evaluate each path: Critically analyze every path—does it align with medical principles? Is
there supporting evidence? Are there logical flaws?
3.Expand the most promising paths: Select 1–2 of the most reasonable paths for deeper
reasoning, integrating strengths from multiple paths when appropriate.
4.Allow backtracking and revision: If a path is disproven upon deeper exploration, explicitly
state this and pivot to alternative branches.
5.Do not assume the provided answer is correct: Completely disregard whether the given
answer is "correct"; derive conclusions solely based on medical logic.
6.Do not reference the source of input paths or answers: Present all reasoning as the
result of autonomous thinking.
##Input Format:
Question: {{question}}
Answer: {{answer}}
Paths: {{paths}}

##Output :
Thought Tree Exploration:
Path 1: [Briefly describe the first reasoning approach]
Path 2: [Briefly describe the second reasoning approach]
Path 3: [Add more paths if necessary, including those you generate yourself]
Path Evaluation:
Analysis of Path 1: [Strengths/weaknesses/medical basis]
Analysis of Path 2: [Strengths/weaknesses/medical basis]
...
Selection of Optimal Path and In-Depth Reasoning: (Choose the path(s) with the strongest
medical grounding and conduct step-by-step, rigorous reaso

Figure 7: Prompt for generating answer with ToT reasoning

> **prompt for evaluate the relevance of knowledge graph paths**
>
> messages = [ {"role": "system", "content": "You are a medical reasoning expert responsible for assigning dual-dimensional relevance scores to each step in a knowledge graph path."}, {"role": "user", "content": """"Question: {question}
>
> Path: {path}
>
> Please assign the following two scores for each hop in the path:
>
> Single-Step Score: Evaluate only the medical plausibility of the current hop (from the current entity to the next entity) on a scale of 1–10.
>
> Multi-Step Score: Evaluate the overall medical coherence and relevance of the sub-path from the start of the path to the target entity of the current hop on a scale of 1–10.
>
> Example:
>
> If the path is ["Diabetes", "Insulin Resistance", "Type 2 Diabetes"], it contains two hops:Hop 1: "Diabetes" → "Insulin Resistance"Hop 2: "Insulin Resistance" → "Type 2 Diabetes"
>
> For Hop 1:Single-Step: Assess whether "Diabetes → Insulin Resistance" is medically reasonable.Multi-Step: Assess the coherence of the sub-path "Diabetes → Insulin Resistance".
>
> For Hop 2:Single-Step: Assess whether "Insulin Resistance → Type 2 Diabetes" is medically reasonable.
>
> Multi-Step: Assess whether the full sub-path "Diabetes → Insulin Resistance → Type 2 Diabetes" forms a coherent and relevant medical pathway.
>
> ```json
> [
>   {
>     "hop_index": 1,"from": "EntityA","to": "EntityB",
>     "single_step_score": 8,"multi_step_score": 8
>   },
>   {
>     "hop_index": 2,"from": "EntityB","to": "EntityC",
>     "single_step_score": 7,"multi_step_score": 8
> }]```"""}
> ]

Figure 8: Prompt for dual-dimensional scoring

```
prompt for  evaluate the relevance of knowledge
              graph paths

messages = [{"role": "system", "content": "As a medical reasoning expert, evaluate
the relevance of knowledge graph paths to medical questions."},
 {"role": "user", "content": """Evaluate the relevance of the following paths to
the question.
QUESTION: {question}
EXPECTED ANSWER: {answer}
Path list:
{paths_text}
### 评分标准:
- 0-3: Low relevance, unlikely to help answer the question
- 4-6: Moderate relevance, may contain some useful information
- 7-10: High relevance, likely to provide a good medical explanation
### output:
Return directly in JSON format, without any other content:
```json
[
  {
    "path_id": 1,"score": 8,"reason": "Brief reason"
  },
  {
    "path_id": 2, "score": 3, "reason": "Brief reason"
  }
]
```"""}
]
```

Figure 9: Prompt for evaluating the relevance of knowledge graph paths

