# OpenReview forum: "TCMReasonSet: A Dataset for Explainable Medical Reasoning in Traditional Chinese Medicine"
_ICLR.cc/2026/Conference — ICLR 2026 Conference Withdrawn Submission_

### Official Review · Reviewer_BnXL · 2025-10-19

**Soundness:** 3
**Presentation:** 3
**Contribution:** 2
**Rating:** 4
**Confidence:** 3

**Summary:**

This paper introduces TCMReasonSet, a dataset tailored for explainable medical reasoning in Traditional Chinese Medicine. The dataset construction involves three core stages: (1) building a TCM knowledge graph with 52,000 entities and 1.38 million relations; (2) generating 50,000 knowledge-constrained QA pairs covering TCM theory, diagnosis, and pharmacology; (3) proposing the TCM Tree-of-Thought (TCM-ToT) methodology with a dual-dimensional scoring mechanism to transform QA pairs into interpretable reasoning chains. Evaluations further validate the dataset’s reliability.

**Strengths:**

1. The paper try to adapt the ToT paradigm to TCM’s distinctive reasoning logic. The integration of a domain-specific knowledge graph into the reasoning pipeline also represents a thoughtful customization for TCM’s complex knowledge structure.
2. The dataset construction exhibits multi-source validation and expert review. The experiments are comprehensive, covering multiple benchmarks and model architectures.
3. The experimental results are clearly organized, enabling easy comparison between models.

**Weaknesses:**

1. The paper relies on existing paradigms (knowledge graphs, ToT, dual-dimensional scoring) and primarily adapts them to TCM rather than proposing entirely new methodologies. For example, the TCM-ToT builds on the original ToT framework, and knowledge-constrained QA generation has been used in other medical subfields (e.g., Western medicine drug discovery).
2. The paper focuses on benchmark performance but provides little insight into how TCMReasonSet or TCMReason-8B would perform in real clinical settings.
3. Critical considerations like inter-expert variability in TCM syndrome differentiation, regulatory compliance for medical AI, and usability for TCM practitioners are not addressed.
4. The experiments focus on model accuracy but lack analysis of reasoning interpretability beyond expert ratings. Additionally, the paper does not compare against non-Chinese TCM reasoning datasets or models, limiting the assessment of its global relevance.
5. The TCM-KG’s entity and relation coverage is not evaluated against other existing TCM knowledge bases (e.g., TCM-SKG), making it hard to assess its comprehensiveness.
6. The dual-dimensional scoring mechanism’s thresholds are not justified, and there is no analysis of how sensitive the results are to these parameters.
7. Most importantly, this paper claims to be open source, but the current version does not include any open source URLs. We recommend that the revised version include an anonymized URL or dataset to further evaluate the dataset.

**Questions:**

1. How did you resolve conflicting information from multi-source data  during TCM-KG construction? Providing specific examples and resolution strategies would strengthen the dataset’s credibility.
2. Providing information about the annotation software or human review process would help improve the feasibility of the article.
3. Currently, the dataset focuses on herbal formulas and syndrome differentiation, limiting its breadth. What is the future work of the current paper?
4. Have you tested the proposed model in real clinical settings with TCM practitioners? If so, what feedback did you receive regarding the model’s reasoning interpretability and clinical utility? If not, what steps will you take to validate its real-world applicability?
5. The dual-dimensional scoring mechanism uses HuatuoGPT-o1-72B as the scoring oracle. How does this choice impact the scoring consistency, and have you considered using multiple scoring models to mitigate potential biases from a single oracle?
6. Why did you not compare TCMReasonSet against existing TCM knowledge graphs or reasoning datasets (e.g., TCMEval-SDT, SymMap-derived datasets)?

---

### Official Review · Reviewer_yJqs · 2025-10-30

**Soundness:** 1
**Presentation:** 1
**Contribution:** 2
**Rating:** 4
**Confidence:** 4

**Summary:**

The paper introduces TCMReasonSet, a reasoning-focused dataset for Traditional Chinese Medicine (TCM). TCMReasonSet is built via 1) a TCM knowledge graph, 2) LLM-generated, knowledge-constrained QA pairs, and 3) a TCM-specific Tree-of-Thought (TCM-ToT) pipeline scored on factual accuracy and global logical coherence. The constructed dataset is used to fine-tune several sub-10B models, which show higher performance gains across various TCM benchmarks when compared to models trained on Huatuo CoT dataset.

**Strengths:**

1. The paper is well motivated based on TCM’s non-linear diagnostic logic and proposes a ToT approach anchored in a KG.
2. Local factuality (LocalScore) and global coherence (CohereScore) are proposed to ensure the data quality, with ablation showing their unique importance.
3. Instruction-tuned and reasoning models fine-tuned on TCMReasonSet show consistent improvements across different TCM benchmarks.

**Weaknesses:**

1. It is unclear why the LLM could not directly generate both the QA pairs and their corresponding reasoning steps simultaneously, given that the QA pairs are derived from paths within the knowledge graph.
2. The paper lacks direct comparisons between TCMReasonSet and Huatuo-CoT on Qwen3-8B and DeepSeek-Distill-8B.
3. Table 4 appears inconsistent with Table 1: the bolded "w/" scores in Table 4 seem to represent the original Qwen2.5-7B-Instruct performance without post-training. According to the table, removing any scoring mechanism leads to results significantly below the Huatuo-CoT-tuned baseline and even the untuned model. This discrepancy requires deeper analysis and clarification.

Minor issues:
- The first listed contribution in the Introduction states "consisting of 30,000 ... ", while other parts of the paper refer to "> 30,000".
- Some tokens in lines 199-202 look weird.
- There is a broken citation for HuatuoGPT-o1-72B in line 303.
- In-text citations with parentheses should be used wherever applicable.

**Questions:**

- Why can't the LLM generate both the QA pairs and the corresponding reasoning steps simultaneously, given that the QA pairs are built from paths within the knowledge graph?
- What is the performance of Huatuo-CoT on Qwen3-8B and DeepSeek-Distill-8B?
- Why does Table 4 show results that appear inconsistent with Table 1, and why does the removal of any scoring mechanism cause significant performance to drop below even the untuned model? Could the authors provide further analyses or clarification for this discrepancy?

---

### Official Review · Reviewer_SE2M · 2025-11-01

**Soundness:** 2
**Presentation:** 3
**Contribution:** 2
**Rating:** 4
**Confidence:** 4

**Summary:**

This paper presents TCMReasonSet, a large-scale dataset designed to enhance explainable reasoning for Traditional Chinese Medicine (TCM). It combines a newly constructed TCM knowledge graph (TCM-KG) with Tree-of-Thought (ToT) reasoning to create structured, interpretable reasoning chains. The work aims to improve the reliability and interpretability of medical large language models, addressing a practical need for transparent clinical reasoning in TCM diagnosis, treatment, and education.

**Strengths:**

1. The paper is well organized and clearly written, with intuitive figures and tables that make the overall pipeline easy to follow.

2. The dataset construction process is well-designed, integrating multiple modules such as KG building, entity alignment, reasoning-chain generation, and multi-stage filtering. This modular pipeline reflects good engineering practice and provides a valuable resource for studying structured medical reasoning.

**Weaknesses:**

1. The entire data construction pipeline relies heavily on large language models rather than natural data linkage or expert curation. Both the KG extraction and the question generation depend on LLM outputs, which could introduce unreliability and systemic bias.

2. The data quality filtering and scoring rely primarily on HuatuoGPT-o1-72B, creating a self-referential setup where one model evaluates data generated by other models. This raises concerns about potential bias and lack of independent validation.

3. The human expert evaluation is limited in scope. The number of evaluated samples is small, the scoring is subjective, and there is no report of inter-rater reliability or consistency, which weakens the credibility of the human assessment.

4. The evaluation design does not fully align with the dataset’s stated goal. While the dataset emphasizes explainable reasoning, most experiments focus on multiple-choice benchmarks. Beyond subjective expert ratings and case studies, there are no objective process-level metrics—such as reasoning consistency, factual accuracy, or path correctness—to quantitatively validate improvements in interpretability.

**Questions:**

1. Since both the TCM-KG construction and QA generation rely heavily on LLM outputs, how do the authors ensure the factual reliability and medical correctness of these automatically extracted or synthesized triples? Was there any human verification or error-rate estimation?

2. How were ambiguous or conflicting knowledge entries handled when the same concept appeared with multiple relations or synonyms across different data sources?

---

### Note · Authors · 2026-01-06

I have read and agree with the venue's withdrawal policy on behalf of myself and my co-authors.